# GaussianMorphing: Mesh-Guided 3D Gaussians for Semantic-Aware Object Morphing

## Abstract

We introduce **GaussianMorphing**, a novel framework for semantic-aware 3D shape and texture morphing from multi-view images. Previous approaches usually rely on point clouds or require pre-defined homeomorphic mappings for untextured data. Our method overcomes these limitations by leveraging mesh-guided 3D Gaussian Splatting (3DGS) for high-fidelity geometry and appearance modeling. The core of our framework is a unified deformation strategy that anchors 3D Gaussians to reconstructed mesh patches, ensuring geometrically consistent transformations while preserving texture fidelity through topology-aware constraints. In parallel, our framework establishes unsupervised semantic correspondence by using the mesh topology as a geometric prior and maintains structural integrity via physically plausible point trajectories. This integrated approach preserves both local detail and global semantic coherence throughout the morphing process without requiring labeled data. On our proposed TexMorph benchmark, GaussianMorphing substantially outperforms prior 2D/3D methods, reducing color consistency error ($\Delta E$) by 22.2% and EI by 26.2%.

## 1 Introduction

Morphing (Gregory et al., 1998; Zhang et al., 2024a) has long been a foundational technique in shape transformation, enabling the generation of continuous interpolation sequences between source and target shapes. Serving as a bridge between computer vision and computer graphics, morphing has emerged as an indispensable tool for applications spanning computer animation, geometric modeling, and shape analysis. Its prominence in visual effects for film and media production further underscores its practical significance.

Existing morphing techniques can be broadly categorized into two paradigms: image-based methods (Aloraibi, 2023; Zhang et al., 2024a) and 3D geometric methods (Eisenberger et al., 2021; Yang et al., 2025; Cao et al., 2024). As summarized in Figure 1, these approaches exhibit fundamental trade-offs. Image-based pipelines, such as DiffMorpher (Zhang et al., 2024b) and FreeMorph (Cao et al., 2025), produce high-fidelity 2D outputs but lack 3D geometric reasoning and multi-view consistency. Extensions like MorphFlow (Tsai et al., 2022) leverage Neural Radiance Fields (NeRF) to address view consistency but are limited by the absence of explicit 3D geometric constraints, resulting in incomplete volumetric reconstructions (denoted as 2.5D* in Figure 1). In contrast, 3D-centric methods such as Neuromorph (Eisenberger et al., 2021) enable mesh-based deformation but require high-quality mesh inputs, neglect texture-aware processing, and struggle with topological complexity. These limitations highlight a critical gap: the lack of a unified framework that balances geometric robustness, textural coherence, and input accessibility without reliance on high-fidelity 3D data, which remains a key challenge for advancing morphing techniques toward practical and general-purpose applications.

To address this gap, this work introduces the first framework for joint 3D geometry and texture morphing using 3D Gaussians, where shape and appearance are intrinsically unified (Figure 1). The key challenge lies in achieving coherent deformation with Gaussian representations due to their unstructured nature and the complexity of maintaining geometry-texture alignment. Our solution integrates the rendering efficiency of 3D Gaussian Splatting (3DGS) (Kerbl et al., 2023) with the structural benefits of mesh-guided deformation. The approach establishes explicit bindings between 3DGS primitives and mesh elements, enabling smooth interpolation while preserving geometric and tex-

| Method | Input Type | Output Type | Texture |
|---|---|---|---|
| DiffMorpher | Images | 2D | ✓ |
| FreeMorph | Images | 2D | ✓ |
| MorphFlow | Images | 2.5D* | ✓ |
| Neuromorph | Mesh | 3D | ✗ |
| GaussianMorphing (Ours) | Images | 3D | ✓ |

Figure 1: Our **GaussianMorphing** (left) takes input images of the source and target, reconstructs them into 3D Gaussian representations with surface meshes, and uses a **mesh-guided strategy** to generate intermediate shapes at timestamps $t \in [0, 1]$. Unlike prior approaches, our method achieves Semantic-Aware Object Morphing with textured colors without relying on 3D input data. The comparison table (right) shows that our method uniquely generates fully textured 3D outputs directly from images, offering complete geometric and textural fidelity.

tural fidelity. Through mesh feature extraction and topological constraints, the method ensures stable morphing sequences that resist the geometric fragmentation typical of discrete point representations. A dual-domain optimization strategy employs geodesic-based geometric distortion loss and texture-aware color smoothness loss to govern deformation, ensuring temporal coherence from accessible 2D inputs without requiring specialized 3D assets.

The proposed framework bridges discrete 3DGS points with semantic-aware mesh structures, achieving significant improvements over state-of-the-art methods in geometric accuracy and texture preservation. Experiments demonstrate robust performance across diverse scenarios, including complex topologies and texture-rich objects, while reducing dependency on high-quality 3D data.

The main contributions are:

(1) A mesh-guided framework that integrates 3D Gaussian Splatting with semantic-aware morphing, enabling high-fidelity 3D interpolation from minimal inputs;

(2) Deformation mechanisms that are aware of both topology and semantics, preventing geometric fragmentation and ensuring stable, coherent morphing in Gaussian-based representations;

(3) A dual-domain optimization strategy combining geodesic-aware geometric constraints and texture-aware color interpolation that achieves seamless visual results.

## 2 RELATED WORK

### 2.1 IMAGE MORPHING

Image morphing is a long-standing problem in computer vision and graphics, aiming to generate smooth and perceptually natural transitions between images (Aloraibi, 2023; Zope & Zope, 2017; Wolberg, 1998). Traditional methods (Beier & Neely, 2023; Bhatt, 2011; Liao et al., 2014) rely on correspondence-driven warping and blending, which preserve visual consistency but struggle with content creation, often leading to artifacts. More recently, optimal transport has been applied to morphing simple 2D geometries (Benamou et al., 2015; Bonneel et al., 2011; Solomon et al., 2015), providing mathematically elegant transformations but lacking the texture richness of natural images. Diffusion-based approaches such as DiffMorpher (Zhang et al., 2024b), AID (He et al., 2024), and FreeMorph (Cao et al., 2025) leverage pre-trained generative models to enable flexible morphing across diverse categories. In this work, we instead start from multi-view inputs, alleviating the need for large-scale pre-training and producing intermediate mesh-based representations that support shape-aware and texture-consistent 3D morphing.

## 2.2 SHAPE MATCHING

The problem of 3D shape correspondence aims to establish point-wise mappings between shapes and has been widely studied. Traditional methods rely on geometric constraints (Holzschuh et al., 2020; Roetzer et al., 2022) or non-rigid registration (Bernard et al., 2020; Eisenberger et al., 2019; Ezuz et al., 2019), but often require costly optimization and manual alignment, limiting scalability. Recent learning-based approaches have advanced the field by training networks to match vertices to a template (Monti et al., 2017; Boscaini et al., 2016; Masci et al., 2015), or by leveraging functional maps with learnable features (Litany et al., 2017; Ovsjanikov et al., 2012). Others integrate spectral and spatial cues (Cao et al., 2024; Attaiki & Ovsjanikov, 2023), use diffusion models for functional map prediction (Zhuravlev et al., 2025), or apply 2D correspondence priors to improve semantic consistency in 3D registration (Liu et al., 2025). Our method, with the assistance of neural networks, eliminates the need for costly 3D inputs and data annotations. By employing object reconstruction techniques, it derives geometric point-wise correspondences from images.

## 2.3 SHAPE INTERPOLATION

Shape interpolation addresses the fundamental challenge of smoothly transforming one shape into another by generating intermediate shapes at specified composition percentages. Traditional geometric methods (Brandt et al., 2016; Heeren et al., 2012; Wirth et al., 2011) formulate this as finding geodesic paths on high-dimensional manifolds, employing deformation metrics like As-Rigid-As-Possible (ARAP) (Sorkine & Alexa, 2007) and PriMo (Botsch et al., 2006) to minimize local distortions. Data-driven approaches alternatively navigate through collections of related shapes (Aydınlılar & Sahillioğlu, 2021; Gao et al., 2017), while physics-based methods model interpolation as constrained gradient flows (Eisenberger & Cremers, 2020; Eisenberger et al., 2019). MorphFlow exemplifies this approach by combining Wasserstein flow with rigidity constraints for multiview morphing (Tsai et al., 2022). Recent neural approaches have advanced unsupervised shape interpolation. NeuroMorph (Eisenberger et al., 2021) and Spectral Meets Spatial (Cao et al., 2024) demonstrate effective frameworks for shape matching and interpolation, with the latter incorporating spectral regularization for handling large non-isometric deformations. Other methods utilize 2D correspondence guidance (Liu et al., 2025) or diffusion priors for textured morphing (Yang et al., 2025). While our method shares the high-level goal of textured 3D morphing from multi-view images with Yang et al., we adopt a fundamentally different technical pathway. Their pioneering generative–regeneration paradigm leverages the imaginative power of a diffusion prior to synthesize intermediate geometry and appearance, whereas our geometrically constrained deformation framework explicitly preserves geodesic distances and local rigidity throughout the morphing process. Our method combines geodesic distance measurements with ARAP constraints, utilizing a neural network-based interpolator to achieve smooth deformation from source to target shapes.

# 3 MESH-GUIDED GAUSSIAN MORPHING

Given source and target objects represented by multi-view images, we propose a semantic-aware 3D morphing framework that addresses a fundamental challenge: achieving geometrically consistent transformations while preserving photorealistic surface details. The core problem is that modern explicit representations present a trade-off: 3D Gaussian Splatting (3DGS) (Kerbl et al., 2023) lacks the topological connectivity needed for structured morphing, while traditional meshes struggle to model complex appearance.

As shown in Figure 2, Our framework, Mesh-Guided Gaussian Morphing, resolves this tension by introducing a novel hybrid paradigm. Our key insight is to impose an explicit triangular mesh as a *topological scaffold* to guide the transformation of unstructured Gaussians. By anchoring Gaussians to this mesh, we can leverage powerful mesh-based correspondence techniques to establish semantic connections. This allows us to compute a geometrically consistent morphing flow in the structured mesh domain while using the rich Gaussian representation for photorealistic rendering at any point in the transformation.

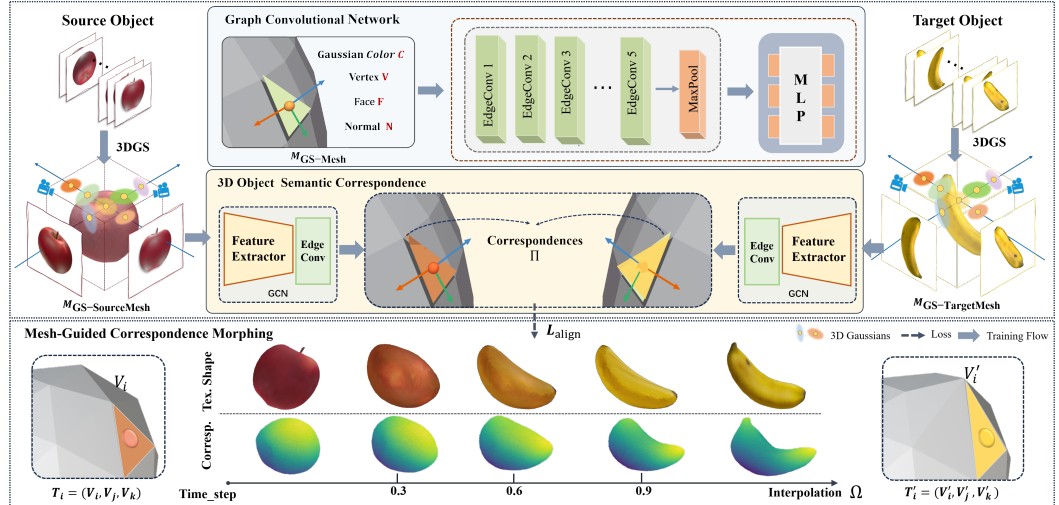

Figure 2: **Method Overview.** Our **GaussianMorphing** framework takes source $\mathcal{X}$ and target $\mathcal{Y}$ images as input. Surface meshes are extracted from 3D Gaussian Splatting (Sec. 3.1) and used with Gaussian points for geometry–texture alignment. Geometric features provide the correspondence matrix $\Pi_{XY}$ (Sec. 3.2), and intermediate shapes are interpolated over time. Training relies on a joint loss (Sec. 3.3), yielding high-quality textured 3D morphing. (Up: Blender results; Down: correspondence visualization with Matplotlib.)

## 3.1 HYBRID MESH-GAUSSIAN REPRESENTATION FOR SEMANTIC MORPHING

**The Connectivity Challenge in Gaussian-Based Morphing.** 3DGS represents a scene as a set of anisotropic 3D Gaussians, which can be optimized to reproduce a set of input images, enabling photorealistic novel-view synthesis. Each Gaussian $g$ is defined by its position $\mu_g \in \mathbb{R}^3$, covariance $\Sigma_g$, opacity $\alpha_g$, and spherical harmonics (SH) coefficients $\mathbf{sh}_g$. While excellent for rendering, the discrete, unstructured nature of these Gaussians prevents the establishment of meaningful semantic correspondences between objects. A direct Gaussian-to-Gaussian matching would likely produce geometrically implausible results that tear or distort the structure of the object.

**Mesh-Anchored Gaussian Binding.** To overcome this limitation, we impose a topological structure by anchoring Gaussians to an explicit mesh. First, we extract a high-quality initial mesh from the optimized Gaussians. We follow recent methods like SuGaR (Guédon & Lepetit, 2024b) and FrostingGaussian (Guédon & Lepetit, 2024a), which use Poisson reconstruction (Kazhdan et al., 2006) alongside regularization terms to ensure the mesh surface accurately reflects the geometry captured by the Gaussians.

With this mesh scaffold, we establish an explicit binding between the Gaussians and the mesh faces. Each Gaussian is anchored to a specific triangular face $f = (V_1, V_2, V_3)$, with its position $\mu_g$ defined by barycentric coordinates $(w_1, w_2, w_3)$ and a normal offset $d$:

$$\mu_g = w_1 V_1 + w_2 V_2 + w_3 V_3 + d \cdot \mathbf{n}_f, \tag{1}$$

where $\mathbf{n}_f$ is the face normal. This binding ensures that as the mesh vertices $V_i$ deform over the course of the morph, the anchored Gaussians move cohesively with the surface, preserving the fine-grained geometric and appearance details they represent.

## 3.2 SEMANTIC CORRESPONDENCE THROUGH TOPOLOGICAL UNDERSTANDING

**Semantic-Aware Mesh Correspondence.** With the mesh structure established, we can tackle the core challenge of identifying which part of the source object should transform into which part of the target. We formulate this as a correspondence problem between the source mesh $(V^S, F^S)$ and target mesh $(V^T, F^T)$. The correspondence is encoded as a probabilistic matrix $\Pi \in \mathbb{R}^{n \times m}$:

$$\Pi_{ij} = P(V_j^T \mid V_i^S) = \frac{\exp(\sigma c_{ij})}{\sum_{k=1}^{m} \exp(\sigma c_{ik})}, \tag{2}$$

where $c_{ij}$ is the cosine similarity between learned feature vectors for source vertex $V_i^S$ and target vertex $V_j^T$. To learn semantically rich features, we use a 5-layer Graph Convolutional Network (GCN) that processes mesh connectivity, allowing it to capture local geometric context without relying on hand-engineered descriptors.

**Neural Morphing Flow.** Rather than simple linear interpolation, we learn a continuous, non-linear deformation field. We employ a neural network, the Correspondence Morphing Flow ($\Psi$), to predict the morphing trajectory. This network ($\Psi$) utilizes the same GCN architecture as our correspondence network, but additionally accepts the time $t$ as an input. This allows $\Psi$ to learn a continuous, non-linear interpolation flow by conditioning its predicted displacement on the time $t \in [0,1]$. At any time $t \in [0,1]$, the morphed source vertices $V^S(t)$ are given by:

$$V^S(t) = V^S + \Psi(V^S, \Pi V^T - V^S, t). \tag{3}$$

Here, the term $\Pi V^T - V^S$ represents the *semantically-aligned displacement field* that maps each source vertex to its corresponding target location. The network $\Psi$ learns to smoothly interpolate this displacement over time.

**Consistent Gaussian Updates.** As the mesh vertices $V^S(t)$ deform, the positions of the bound Gaussians $\mu_g(t)$ are updated consistently via the barycentric relationship established in Eq. 1:

$$\mu_g(t) = \sum_{i=1}^{3} w_i V_{f_i}(t), \tag{4}$$

where $V_{f_i}(t)$ are the deformed positions of the vertices of the triangle $f$ to which Gaussian $g$ is bound. This maintains the tight coupling between the mesh and the Gaussians throughout the entire morphing sequence.

### 3.3 MULTI-OBJECTIVE OPTIMIZATION FOR PLAUSIBLE MORPHING

We optimize the correspondence matrix $\Pi$ and the morphing flow network $\Psi$ using a comprehensive loss function that balances geometric structure, appearance consistency, and semantic alignment.

**Geometric Consistency.** To prevent unnatural stretching and distortion, we enforce that the intrinsic geometric structure of the surfaces is preserved. We measure this using geodesic distances on the mesh. To compute the geodesic distance $D_g(i,j)$ between any two vertices, we run Dijkstra's algorithm on a hybrid graph formed by the union of the mesh adjacency graph $G_{adj}$ (preserving topology) and a KNN graph $G_{knn}$ (adding shortcuts to better approximate Euclidean distances). Further details are provided in Appendix A.2. The geodesic distortion loss is then:

$$\mathcal{L}_{geo} = \left\| \Pi D_g^T \Pi^\top - D_g^S \right\|_F^2, \tag{5}$$

where $D_g^S$ and $D_g^T$ are the geodesic distance matrices for the source and target meshes, and $\|\cdot\|_F$ is the Frobenius norm. This loss encourages the correspondence $\Pi$ to map regions of the target mesh back to the source mesh in a way that respects their intrinsic geometry.

To further encourage local rigidity, we add an As-Rigid-As-Possible (ARAP) energy term (Sorkine & Alexa, 2007), which penalizes non-rigid deformations. We evaluate this over sampled timesteps during the morph:

$$\mathcal{L}_{arap} = \mathbb{E}_{t \sim U[0,1]} \left[ E_{arap}(\mathbf{X}(t), \mathbf{X}(t + \delta t)) \right], \tag{6}$$

where $\mathbf{X}(t)$ is the mesh state at time $t$ and $\delta t$ is a small perturbation.

**Appearance Consistency.** To ensure smooth visual transitions, we introduce a geodesic-aware smoothness loss on the vertex colors. We first initialize the color of each vertex by averaging the RGB colors of its bound Gaussians (with SH coefficients evaluated from a canonical viewing direction). The loss then penalizes color differences between adjacent vertices, weighted inversely by their geodesic distance:

$$\mathcal{L}_{smooth} = \sum_{(i,j) \in E_{adj}} \frac{1}{D_g(i,j) + \epsilon} \cdot \left\| C_{morph}^i(t) - C_{morph}^j(t) \right\|_2^2, \tag{7}$$

where $E_{adj}$ is the set of edges in the mesh adjacency graph. The smoothing loss incorporates a small constant $\epsilon = 1e\text{-}5$ to ensure numerical stability during optimization. This encourages smooth color fields while allowing for sharp transitions across distant parts of the object.

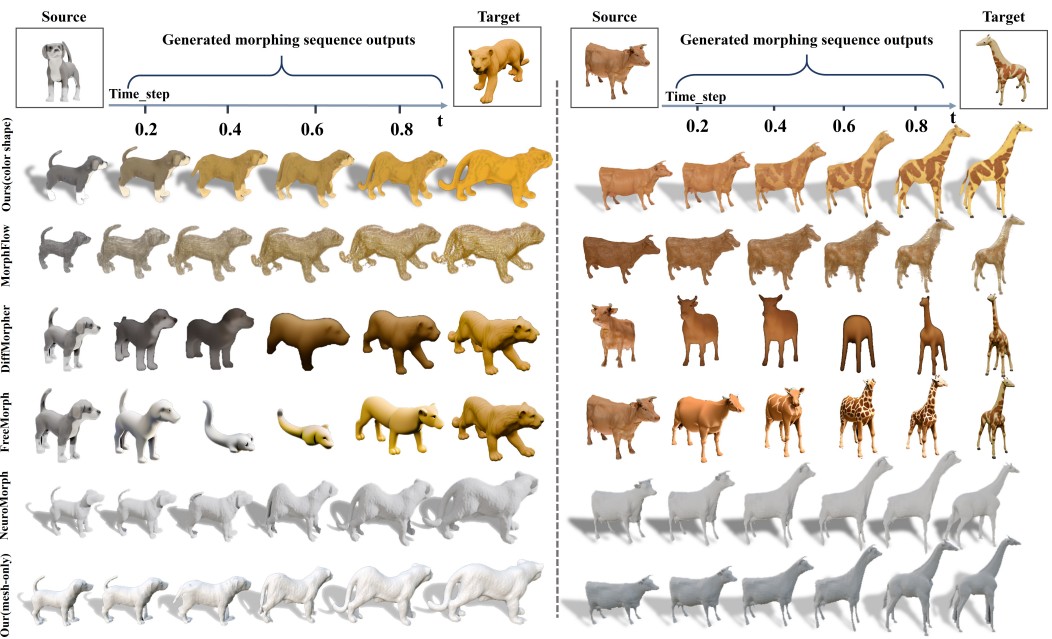

Figure 3: Qualitative comparison of morphing methods on the benchmark dataset. Baselines include DiffMorpher (Zhang et al., 2024b) and FreeMorph (Cao et al., 2025) for image morphing, Neuro-Morph (Eisenberger et al., 2021) for texture-free 3D shape morphing, and MorphFlow (Tsai et al., 2022) for textured multi-view morphing without true geometry. Our method generates textured 3D morphing with geometric details directly from image inputs.

**Semantic Alignment Constraint.** To ensure the morphing sequence reaches its destination, we add a terminal constraint that drives the deformed source mesh to the target configuration at the final timestep:

$$\mathcal{L}_{\text{align}} = \left\| V^S(t=1) - \Pi V^T \right\|_F^2. \tag{8}$$

This loss acts as a boundary condition, ensuring that the morph respects the learned semantic correspondences.

**Unified Loss Function.** Our final objective function is a weighted sum of these components:

$$\mathcal{L}_{\text{total}} = \lambda_{\text{geo}}\mathcal{L}_{\text{geo}} + \lambda_{\text{arap}}\mathcal{L}_{\text{arap}} + \lambda_{\text{smooth}}\mathcal{L}_{\text{smooth}} + \lambda_{\text{align}}\mathcal{L}_{\text{align}}, \tag{9}$$

where the $\lambda$ hyperparameters balance the competing objectives of geometric fidelity, structural rigidity, appearance consistency, and semantic alignment.

## 4 EXPERIMENTS

We conduct comprehensive experiments to validate the ability of our method to produce high-quality, semantically consistent 3D morphs. We introduce a new benchmark, **TexMorph**, designed specifically for this task. Our evaluation protocol includes quantitative comparisons against state-of-the-art 2D and 3D methods using novel metrics, qualitative analysis of the generated morphing sequences, and an ablation study to analyze the contributions of our proposed key components.

### 4.1 EXPERIMENTAL SETUP

**TexMorph Benchmark.** To rigorously evaluate 3D morphing from multi-view images, we created a new benchmark named **TexMorph** (**Tex**ture-rich, **M**orphing-focused). The benchmark is comprised of challenging source-target pairs designed to test geometric and appearance transformations. It includes: (1) high-fidelity synthetic models with complex textures rendered from multiple viewpoints; (2) real-world objects captured via 3D scanning; and (3) objects captured in-the-wild using

Figure 4: Qualitative morphing results with non-isometric deformations, demonstrating robust interpolation under challenging geometric conditions (Up: synthetic datas; Middle: real-world scanned objects from GSO (Downs et al., 2022); Bottom: real-world photos).

standard mobile phone cameras. The dataset features over ten object categories, including animals, fruits, and vehicles, providing diverse topological and textural challenges. Further details are provided in Appendix A.1.

**Evaluation Metrics.** Standard metrics for novel-view synthesis are inadequate for evaluating morphing sequences. We thus propose three metrics to assess the spatio-temporal quality of the transformation from source ($t = 0$) to target ($t = 1$):

- **Structural Stability (MSE-SSIM):** Measures geometric consistency by computing the Mean Squared Error of the temporal SSIM scores against an ideal linear trajectory.

$$\mathcal{E} = \frac{1}{N} \sum_{t \in T} \Big( \text{SSIM}_{\text{ideal}}(A, G_t) - \text{SSIM}_{\text{actual}}(A, G_t) \Big)^2 + \Big( \text{SSIM}_{\text{ideal}}(G_t, B) - \text{SSIM}_{\text{actual}}(G_t, B) \Big)^2.$$
(10)

  A lower value indicates a more stable transformation with fewer structural artifacts.

- **Color Consistency ($\Delta E$):** Assesses appearance smoothness by averaging the perceptual color difference ($\Delta E_{ab}^*$) between corresponding surface points throughout the morph.

$$\Delta E_{ab}^* = \sqrt{(L_1^* - L_2^*)^2 + (a_1^* - a_2^*)^2 + (b_1^* - b_2^*)^2}.$$
(11)

  A lower $\Delta E$ signifies a smoother transition without color bleeding.

- **Edge Integrity (EI):** Evaluates silhouette continuity by measuring the temporal stability of the rendered edge map of object.

$$EI = N_{Edges}(Canny(I, T_{low}, T_{high})) - 1.$$
(12)

  A lower score indicates less fragmented edges, suggesting more stable structural transition in the morphing sequence.

Detailed formulations are available in Appendix A.3.

**Implementation Details.** All experiments were conducted on a single NVIDIA RTX A6000 GPU. For a typical object pair with a mesh of approximately 12,000 faces, the initial hybrid mesh-Gaussian representation is generated in about 1 hour. The optimization of our morphing framework takes between 500 and 1000 iterations, depending on mesh complexity. Once trained, generating a full, high-resolution morphing sequence takes approximately 2 minutes.

## 4.2 EVALUATION

We perform a comprehensive evaluation of GaussianMorphing against several state-of-the-art 2D and 3D morphing methods. For 2D baselines, we compare against DiffMorpher (Zhang et al., 2024b), a diffusion-based method, and FreeMorph (Cao et al., 2025), a tuning-free approach. For 3D baselines, we include MorphFlow (Tsai et al., 2022), which leverages optimal transport for multi-view transitions, and NeuroMorph (Eisenberger et al., 2021), which computes topology-aligned shape correspondences.

| Method | MSE(SSIM)↓ | $\Delta E\downarrow$ | EI ↓ |
|--------|-----------|------|------|
| DiffMorpher | 0.19 | 105 | 97 |
| MorphFlow | 0.17 | 8.23 | 33.6 |
| Neuromorph | 0.13 | / | 13.0 |
| FreeMorph | 0.20 | 13.0 | 21.6 |
| Our | **0.11** | **6.40** | **9.0** |

Table 1: Quantitative comparison of morphing methods evaluates structural similarity using the MSE of SSIM, color consistency with $\Delta E$, and edge continuity through EI.

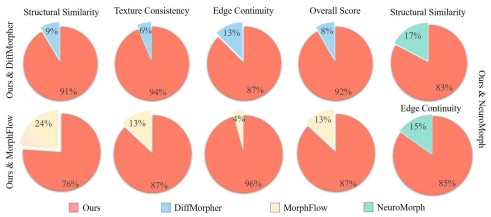

Figure 5: User study results: Comparing our method with the baseline methods in terms of color consistency, structural similarity, and edge continuity. A higher percentage of participants preferred our results across all metrics.

**Textured Morphing Analysis.** Our method excels at producing smooth, high-fidelity texture transitions, as qualitatively demonstrated in Figure 3. The linear color interpolation of MorphFlow is inadequate for high-dimensional color spaces, leading to oversmoothed transitions and loss of detail. For example, during the "dog→lion" transformation, it reduces the morph to a simple color shift, failing to preserve the intricate fur patterns of the lion or the distinct white patches of the dog. The 2D methods perform poorly on challenging cross-category pairs; DiffMorpher fails in both geometric and color alignment, while the 2D SOTA, FreeMorph, introduces severe structural artifacts (*e.g.*, lizard-like textures) and color oversaturation. In contrast, our approach achieves superior color fidelity, corroborated by lower $\Delta E$ values (Table 1), and maintains fine-grained texture details throughout the transformation. Furthermore, as shown in Figure 4, our method produces smooth and plausible morphing results even in the presence of significant non-isometric deformations. The interpolated sequences remain visually coherent, demonstrating the robustness of our approach under challenging geometric conditions. By covering synthetic models, real-world scanned objects from GSO (Downs et al., 2022), and photographs of everyday items, the results further highlight the generalization ability of Gaussian morphing across diverse data sources.

**Geometric and Structural Analysis.** As shown in Table 1, our method achieves state-of-the-art structural consistency and edge continuity, primarily due to $\mathcal{L}_{geo}$, which preserves local geometric details. For a fair comparison with NeuroMorph, we use the same input meshes for both methods. NeuroMorph relies on mesh connectivity for geodesic computation making it brittle when handling fragmented or coarse geometries. Our hybrid graph representation bypasses this dependency, yielding a more robust and efficient solution. Furthermore, our semantic-aware mechanism produces more plausible deformations, correctly preserving features like the tail in "dog→lion" morphs and neck details in giraffe morphs, where NeuroMorph falters. MorphFlow suffers from a lack of constraints on mesh topology or semantic information, an absence that leads to noticeable edge fragmentation and silhouette tearing. By contrast, our topology-aware framework effectively avoids these issues by leveraging the mesh structure to ensure enhanced edge continuity.

**User Study.** To validate the perceptual quality of our results, we conducted a user study with 54 participants, who compared outputs of our method against those from DiffMorpher, MorphFlow, NeuroMorph and the Ablation study. The evaluation focused on four criteria: structural similarity, texture consistency, edge continuity, and overall preference.The criteria shown below:

- **Structural Similarity**: Preservation of structure in intermediate frames.
- **Texture Consistency**: Smooth and natural color transitions without abrupt jumps.
- **Edge Continuity**: Smooth and continuous edges without breaks or distortions.
- **Overall Score**: Comprehensive evaluation based on structural similarity, texture consistency, and edge continuity.

Full details are provided in Appendix A.5. The results show an overwhelming preference for our method across all metrics. Over $80\%$ of users rated our morphs as superior overall, with particularly strong and consistent agreement on aspects such as texture consistency and edge continuity. This perceptual validation confirms that our method generates more visually coherent and high-quality morphs, aligning with our quantitative experiments.

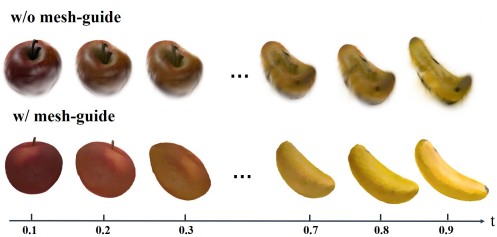

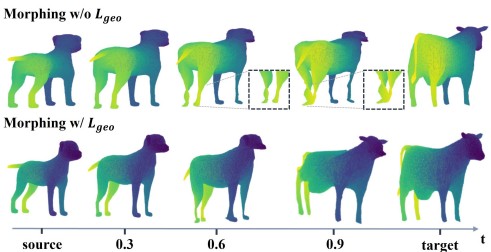

Figure 6: Ablation Study for mesh-guided strategy. Top: Morphing without the **mesh-guided strategy**. Bottom: Morphing with the strategy, demonstrating its role in achieving edge-continuous and smooth transitions.

Figure 7: Ablation Study for **geometric distortion loss**. Comparison of morphing results without (up) and with (below) the geometric distortion loss.

## 4.3 ABLATION STUDY

We conducted an ablation study to isolate the contributions of our core components: the mesh-guided strategy and the geometric distortion loss.

**Importance of Mesh Guidance.**

To evaluate the critical role of mesh guidance in maintaining morphing coherence, we first conducted a comparative evaluation between two distinct approaches: (1) a variant of our method that removes mesh guidance, relying solely on point-based morphing, and (2) our full model, which incorporates mesh guidance by leveraging the complete mesh topology (including vertices, edges, faces, and normals) to establish a shared correspondence $\Pi$. As summarized in Table 2 and illustrated in Figure 6, the point-based variant fails to maintain structural coherence, resulting in significant tearing and discontinuities along object surfaces, particularly noticeable at edges. Quantitatively, this structural degradation is reflected in a higher Edge Continuity Index (EI) score of 34.3, indicating poorer performance. The absence of topological guidance also compromises texture quality, leading to blurry artifacts. In contrast, the mesh-guided approach effectively preserves structural integrity and ensures smoother transitions by enforcing spatial and textural consistency through the explicit use of mesh structure.

**Role of Geometric Distortion Loss.**

Next, we ablate the geometric distortion loss. Without this constraint, the morphing process introduces severe and unnatural deformations, such as the distorted leg geometry shown in Figure 7. These artifacts not only degrade visual quality but also disrupt the structural plausibility of the interpolated shapes, making the transitions appear unrealistic. By explicitly penalizing local shape changes, this loss serves as a key regularizer that

Table 2: Mesh-Guided Strategy Ablation: Quantifying edge continuity (EI), user-rated transition quality, and texture preservation (MSE(SSIM)) to validate the importance of mesh guidance for smooth shape and texture morphing.

|  | Edge Continuity | | Texture Quality |
| --- | --- | --- | --- |
|  | EI↓ | User↑ | MSE(SSIM)↓ |
| w/o Mesh-Guided | 34.3 | 0.02 | 0.34 |
| w/o $\mathcal{L}_{smooth}$ | – | – | 0.22 |
| Ours | **9.0** | **0.98** | **0.11** |

preserves structural integrity, enforces geometric continuity, and produces smoother, more plausible transformations. User feedback corroborates this finding, confirming a marked reduction in visual distortion when the loss is applied.

**Impact of ARAP Loss on Structural Integrity**

To verify the contribution of the As-Rigid-As-Possible (ARAP) constraint, we visualize the morphing sequence with this term disabled. Fundamentally, the ARAP loss serves as a temporal regularizer, computed between adjacent frames ($t$ and $t + \delta t$) throughout the optimization process. By penalizing non-rigid deformations across these consecutive time steps, it enforces the preservation of local geometric features, ensuring that the mesh triangles undergo physically plausible rotations and translations rather than arbitrary scaling or shearing.

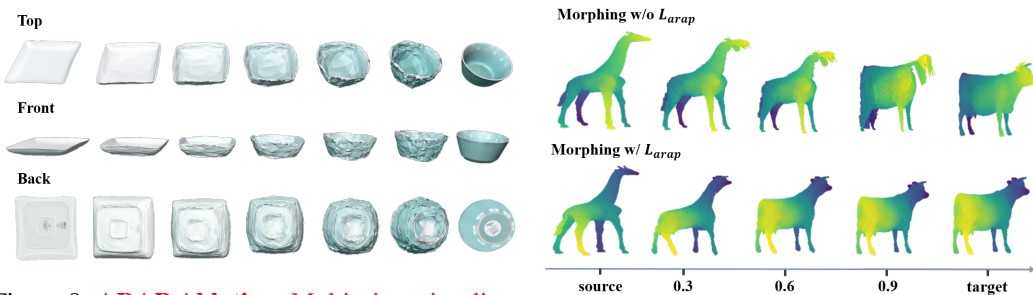

Figure 8: **ARAP Ablation.** Multi-view visualization of the "plate-to-bowl" case with the ARAP loss excluded, shown from three different perspectives.

Figure 9: Ablation Study for **ARAP loss**. Comparison of morphing results without (up) and with (below) the ARAP loss.

As observed in our experiments (see in Figure 8 and Figure 9), removing this constraint leads to unnatural distortions; specifically, the "plate-to-bowl" case suffers from a loss of structural integrity, while the "giraffe-to-cow" sequence exhibits severe edge twisting and arbitrary stretching.

In summary, these studies demonstrate that the synergy between mesh guidance, geometric distortion loss and the ARAP loss is essential for achieving high-fidelity geometric and textural transformations, significantly improving geometric continuity and leading to more natural morphing results.

## 5 CONCLUSION

We introduced **GaussianMorphing**, a novel semantic-aware framework that unifies 3D shape and texture morphing from multi-view images. Our key innovation is a *mesh-guided Gaussian morphing* strategy that anchors 3D Gaussians to semantic mesh patches. This approach bypasses the need for pre-aligned 3D assets and ensures that geometry and appearance are interpolated in a structurally consistent and texturally coherent manner. Through unsupervised learning guided by mesh topology, our method achieves state-of-the-art performance, outperforming existing 2D and 3D techniques in structural similarity, color consistency, and edge continuity. By generating efficient and visually faithful transformations, GaussianMorphing sets a new standard for 3D morphing and opens up new possibilities for applications in visual effects and digital content creation.

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

# A APPENDIX

**Statements** The language of this manuscript was refined with the assistance of a large language model (LLM). The authors remain fully responsible for the originality, accuracy, and integrity of all academic content, analyses, and ideas presented in the paper.

## A.1 TEXMORPH BENCHMARK

Our new morphing benchmark, TexMorph, leverages high-precision synthetic object models crafted by artists and 3D object models captured from real scenes, forming a diverse dataset that spans multiple object categories, all constructed in Blender.

We utilize the $BlenderNeRF$ plugin to define a spherical orbit path for an active camera($COS$) around the object. Training frames are rendered by uniformly sampling random camera views oriented toward the center. The dataset includes over ten categories of objects, such as synthetic and real-world collected (scanning model and photo) fruits, animals, furniture, vehicles, and more(see in Figure 10). We utilize this benchmark to conduct both qualitative and quantitative tests on the baselines mentioned below, evaluating the superiority of our method.

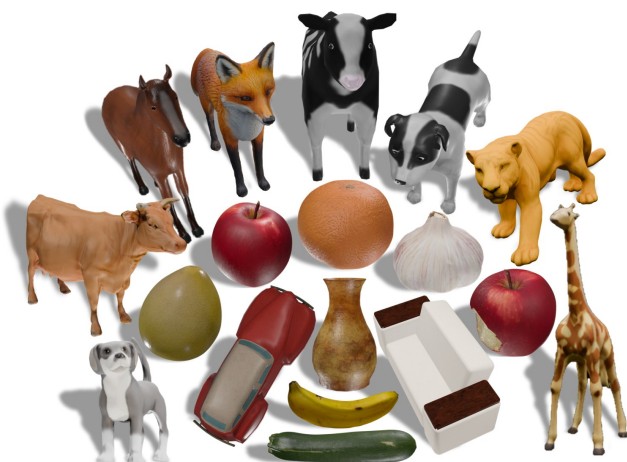

Figure 10: Examples of objects from the texmorph dataset, emphasizing the diversity in texture and structural features for morphing assessment

## A.2 GEODESIC DISTANCE APPROXIMATION

Computing exact geodesic distances on meshes is computationally expensive for large-scale morphing. We approximate them using a hybrid graph representation that balances accuracy and efficiency. We construct two complementary graphs: the adjacency graph $G_{\text{adj}}$ ensures topological consistency by connecting face-adjacent vertices, while the KNN graph $G_{\text{knn}}$ provides local geometric awareness for improved distance approximation in sparse regions. The adjacency graph $G_{\text{adj}}$ encodes face-sharing connectivity:

$$G_{\text{adj}} = \big\{(v_i, v_j) \mid v_i, v_j \text{ share a face in } F\big\}. \tag{13}$$

The KNN graph $G_{\text{knn}}$ captures local Euclidean proximity:

$$G_{\text{knn}} = \big\{(v_i, v_j) \mid d(v_i, v_j) \leq \text{NN-distance}, i \neq j\big\}, \tag{14}$$

where $d(\cdot, \cdot)$ denotes Euclidean distance, and $k = 500$ defines the number of nearest neighbors.

Combining $G_{\text{adj}}$ and $G_{\text{knn}}$, we construct a hybrid distance matrix $D_{\text{adj}} \in \mathbb{R}^{n \times n}$:

$$D_{\text{adj}}(i,j) = \begin{cases} d(v_i, v_j), & \text{if } (v_i, v_j) \in G_{\text{adj}} \cup G_{\text{knn}}, \\ \infty, & \text{otherwise.} \end{cases} \tag{15}$$

The geodesic distance $D_{\mathrm{g}}(i, j)$ between vertices $v_i$ and $v_j$ is then computed via Dijkstra's algorithm:

$$D_{\mathrm{g}}(i, j) = \min_{P \in \mathcal{P}(v_i, v_j)} \sum_{(v_k, v_{k+1}) \in P} D_{\mathrm{adj}}(v_k, v_{k+1}), \tag{16}$$

where $\mathcal{P}(v_i, v_j)$ is the set of all paths between $v_i$ and $v_j$.

### A.3 EXPERIMENT METRIC DETAILS

For structural similarity, Structural Similarity Index (SSIM) measures the similarity in shape and structure between the Morphing result and the target,we use MSE (SSIM) to measure the deviation between the actual SSIM curve and the ideal linear curve. For color consistency, $\Delta E$ is used to measure the color difference between the source object and target object, ensuring consistency in color. Finally, for edge continuity, Edge Integrity (EI) evaluates the continuity and completeness of edges during the shape morphing process, ensuring that the generated structures maintain consistent and unbroken boundaries.

**SSIM for Structural Similarity**

To ensure smooth and natural shape transitions during 3D morphing, we measure the Structural Similarity Index (SSIM) variation across different morphing stages. Ideally, SSIM should change linearly from the source shape $A$ to the target shape $B$.

In an ideal scenario, we define the expected SSIM values at any morphing stage $t$ as follows:

$$SSIM_{\mathrm{ideal}}(A, G_t) = 1 - t, \tag{17}$$

$$SSIM_{\mathrm{ideal}}(G_t, B) = t, \tag{18}$$

where $G_t$ represents the intermediate shape at stage $t$. This ensures a smooth, gradual transition from $A$ to $B$. For example, at specific morphing stages (At 30%): $SSIM(A, G_{30\%}) = 0.7$, $SSIM(G_{30\%}, B) = 0.3$.

To quantify how closely the actual SSIM values follow the ideal linear transition, we compute the Mean Squared Error (MSE) for each stage $t$:

$$\mathcal{E} = \frac{1}{N} \sum_{t \in T} \Big( SSIM_{\mathrm{ideal}}(A, G_t) - SSIM_{\mathrm{actual}}(A, G_t) \Big)^2 + \Big( SSIM_{\mathrm{ideal}}(G_t, B) - SSIM_{\mathrm{actual}}(G_t, B) \Big)^2 \tag{19}$$

A smaller error indicates that the SSIM variation is nearly linear, reflecting high-quality 3D morphing with smooth transitions and minimal distortion, whereas a larger error suggests anomalous SSIM changes, potentially indicating irregularities or distortions in the 3D morphing process.

**$\Delta E$ for Color Consistenc**

We evaluate color consistency using the $\Delta E$ metric in CIELAB space, calculating the average $\Delta E$ for each frame against the source, target, and adjacent frames. The CIELAB color space is chosen for its perceptual uniformity, where Euclidean distances correspond more closely to human color perception compared to RGB space.

The $\Delta E$ metric quantifies the perceptual difference between two colors and is defined as:

$$\Delta E_{ab}^* = \sqrt{(L_1^* - L_2^*)^2 + (a_1^* - a_2^*)^2 + (b_1^* - b_2^*)^2}, \tag{20}$$

where $L^*$ represents lightness (0-100), $a^*$ represents the green-red axis, and $b^*$ represents the blue-yellow axis in CIELAB space.

For morphing evaluation, we compute three types of color consistency metrics: **source consistency** ($\Delta E_{source}$) measuring deviation from the source image, **target consistency** ($\Delta E_{target}$) evaluating progression toward the target, and **temporal consistency** ($\Delta E_{diff}$) assessing smoothness between consecutive frames.

The final color consistency score is computed as:

$$\Delta E_{avg} = \frac{1}{3}(\bar{\Delta E}source + \bar{\Delta E}target + \bar{\Delta E}_{diff}). \tag{21}$$

Lower $\Delta E_{avg}$ values indicate better color consistency throughout the morphing sequence.

**EI for Edge Continuity**

Edge Integrity (EI) quantifies edge fragmentation by counting connected edge components after Canny edge detection. This metric evaluates structural quality and object boundary preservation in morphed images. EI is computed as:

$$EI = N_{Edges}(Canny(I, T_{low}, T_{high})) - 1. \tag{22}$$

where $N_{Edges}$ represents the number of connected edge components, and the subtraction of 1 excludes the background component. Higher EI values indicate more fragmented edges, suggesting potential structural artifacts in the morphing sequence.

## A.4 PSEUDOCODE

---
**Algorithm 1** GaussianMorphing Algorithm

---
**Require:** Multi-view images for Source ($I_S$) and Target ($I_T$)
                  ▷ 1. Initialization
1:  $G_S, G_T \leftarrow$ Reconstruct3DGS($I_S, I_T$)            ▷ Build Gaussians
2:  $M_S(V^S, F^S), M_T(V^T, F^T) \leftarrow$ ExtractMesh($G_S, G_T$)    ▷ Extract meshes
3:  $B_S \leftarrow$ BindGaussiansToMesh($G_S, M_S$)          ▷ Anchor $G_S$ to $M_S$
4:  Initialize GCN $f_\theta$, MLP $\Psi_\phi$             ▷ Init networks
                  ▷ 2. Optimization Loop
5:  **while** not converged **do**
6:     $\Pi \leftarrow$ ComputeCorrespondence($f_\theta(M_S), f_\theta(M_T)$)    ▷ Get correspondence
7:     $t \sim U[0, 1]$                ▷ Sample time $t$
8:     $V^S(t) \leftarrow V^S + \Psi_\phi(V^S, \Pi V^T - V^S, t)$      ▷ Deform $M_S$
9:     $\mathcal{L}_{geo} \leftarrow ||\Pi GeoDist(M_T)\Pi^\top - GeoDist(M_S)||_F^2$    ▷ Geom. loss
10:    $\mathcal{L}_{arap} \leftarrow E_{arap}(V^S(t))$           ▷ Rigidity loss
11:    $\mathcal{L}_{smooth} \leftarrow$ ColorSmoothnessLoss($G_S, G_T, \Pi, t, M_S$)  ▷ Appearance loss
12:    $\mathcal{L}_{align} \leftarrow ||(V^S + \Psi_\phi(\ldots, 1)) - \Pi V^T||_F^2$    ▷ Alignment loss
13:    $\mathcal{L}_{total} \leftarrow \lambda_{geo}\mathcal{L}_{geo} + \lambda_{arap}\mathcal{L}_{arap} + \lambda_{smooth}\mathcal{L}_{smooth} + \lambda_{align}\mathcal{L}_{align}$
14:    Update $\theta, \phi$ using $\nabla\mathcal{L}_{total}$          ▷ Backpropagate
15: **end while**
                  ▷ 3. Morphing Function
16: **function** GENERATEMORPHING($(t, V^S, V^T, \Pi, \Psi_\phi, B_S)$)
17:    $V^S(t) \leftarrow V^S + \Psi_\phi(V^S, \Pi V^T - V^S, t)$    ▷ Get deformed vertices
18:    $G_S(t) \leftarrow \{\}$             ▷ Init new Gaussian set
19:    **for** each Gaussian $g \in G_S$ **do**
20:       $(f, \boldsymbol{w}) \leftarrow B_S[g]$          ▷ Get binding
21:       $\mu_g(t) \leftarrow \boldsymbol{w} \cdot V^S(t)[f]$      ▷ Update position
22:       $\ldots$             ▷ Interpolate attributes
23:       $G_S(t) \leftarrow G_S(t) \cup \{g(t)\}$
24:    **end for**
25:    **return** $G_S(t)$          ▷ Return morphed Gaussians
26: **end function**

---

## A.5 USER STUDY

To evaluate the 3D morphing quality from a human perspective, we conducted a user study with 54 participants. Each participant viewed 17 questions on multiple pairs of objects, randomly selected from our method and three baseline techniques, to evaluate texture and geometric shape comparisons, as well as the ablation results of our mesh-guided strategy and geometric distortion loss. They

were asked to select the best set of results based on the following criteria: structural similarity, color consistency, edge continuity, and overall quality. The questionnaire used in our user study, designed to evaluate the quality and effectiveness of the morphing results, is shown in the Figure below:

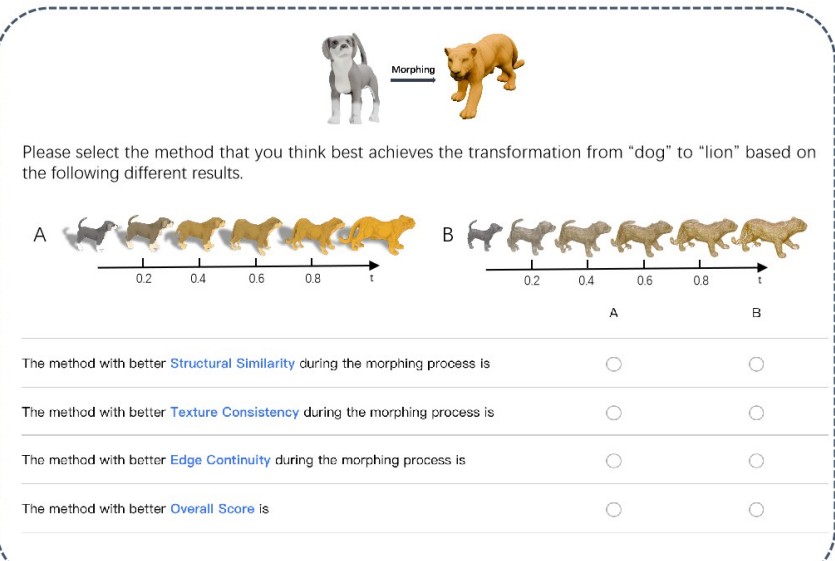

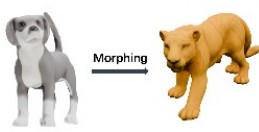

Please select the method that you think best achieves the transformation from "dog" to "lion" based on the following different results.

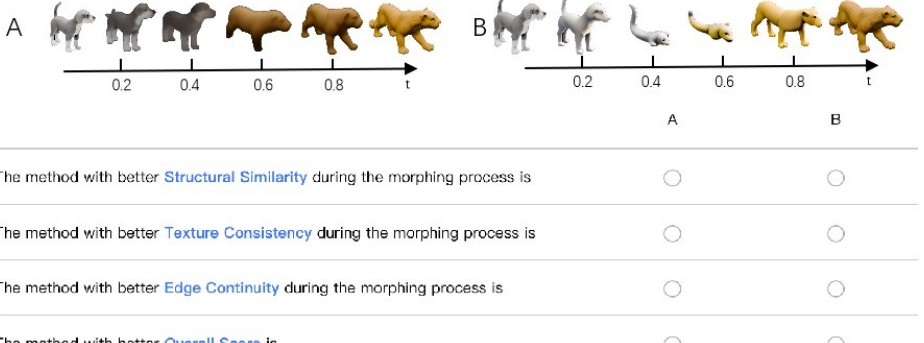

|  | A | B |
| --- | --- | --- |
| The method with better Structural Similarity during the morphing process is | ○ | ○ |
| The method with better Texture Consistency during the morphing process is | ○ | ○ |
| The method with better Edge Continuity during the morphing process is | ○ | ○ |
| The method with better Overall Score is | ○ | ○ |

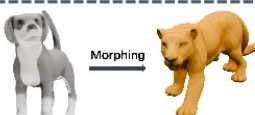

Please select the method that you think best achieves the transformation from "dog" to "lion" based on the following different results.

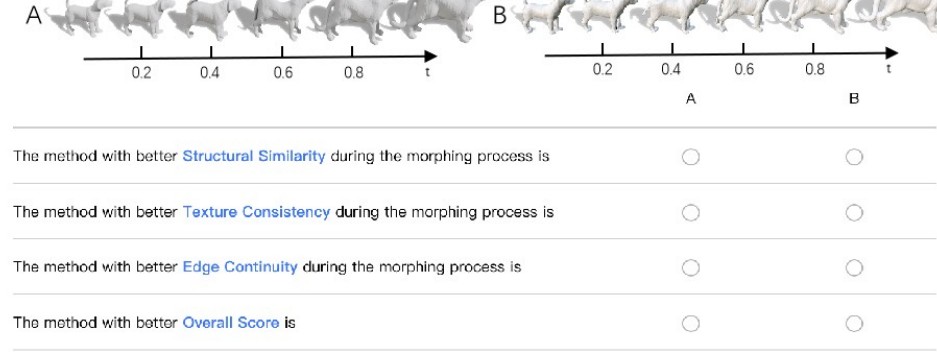

|  | A | B |
| --- | --- | --- |
| The method with better Structural Similarity during the morphing process is | ○ | ○ |
| The method with better Texture Consistency during the morphing process is | ○ | ○ |
| The method with better Edge Continuity during the morphing process is | ○ | ○ |
| The method with better Overall Score is | ○ | ○ |

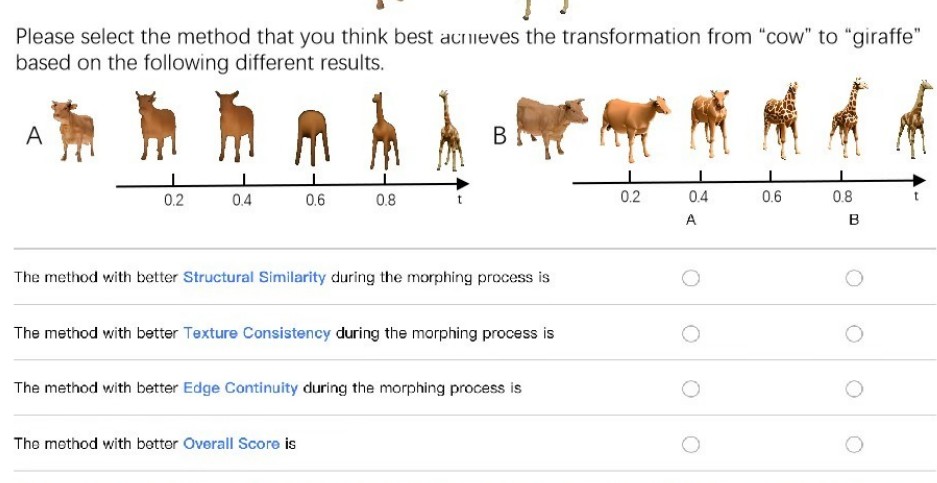

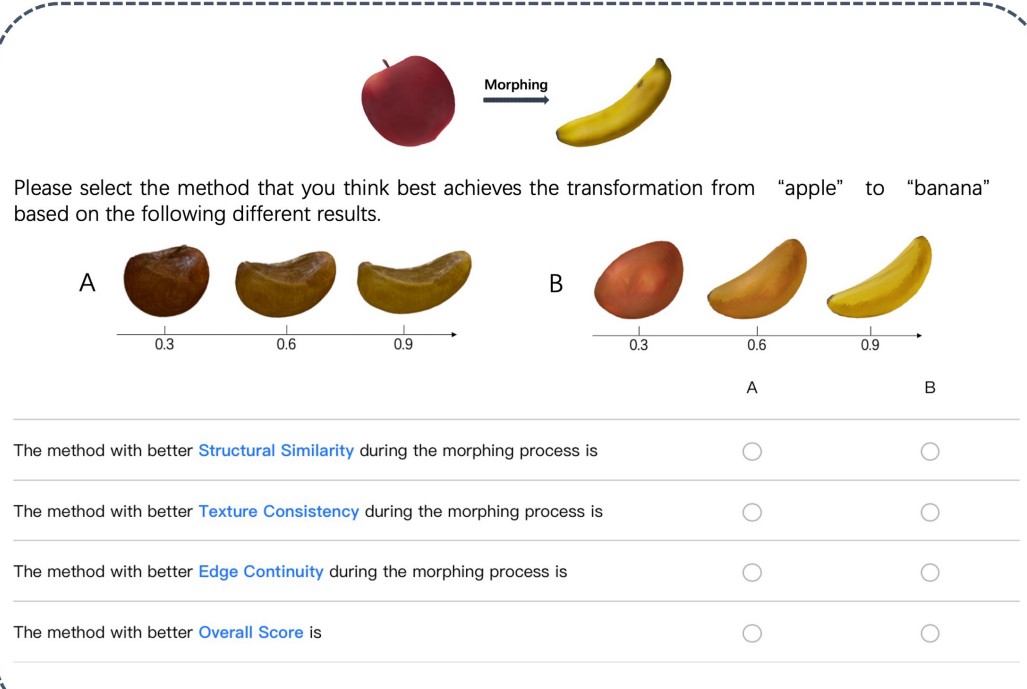

Please select the method that you think best achieves the transformation from "cow" to "giraffe" based on the following different results.

| | A | B |
|---|---|---|
| The method with better Structural Similarity during the morphing process is | ○ | ○ |
| The method with better Texture Consistency during the morphing process is | ○ | ○ |
| The method with better Edge Continuity during the morphing process is | ○ | ○ |
| The method with better Overall Score is | ○ | ○ |

The two images below show the 90% transformation of an apple into a banana using different methods. Which one do you think has better results?

The reason you think this image has better results is...

☐ Structural stability

☐ Edge continuity

☐ Smooth color

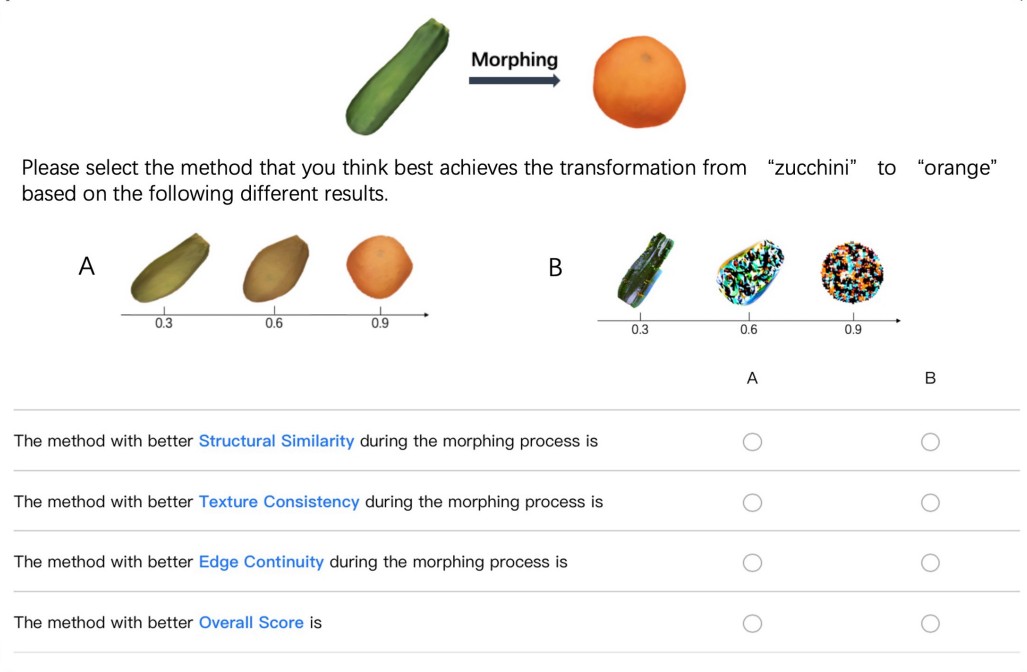

