# OpenReview forum: "GaussianMorphing：Mesh-Guided 3D Gaussians for Semantic-Aware Object Morphing"
_ICLR.cc/2026/Conference — Submitted to ICLR 2026_

### Official Review · Reviewer_ib2M · 2025-10-18

**Soundness:** 3
**Presentation:** 3
**Contribution:** 3
**Rating:** 4
**Confidence:** 3

**Summary:**

The paper addresses the task of semantic-aware 3D shape and texture morphing between source and target objects given only multi-view images, without requiring pre-aligned 3D models or labeled data. The core technical pipelines involves 1) reconstructing 3DGS representations from the input images, 2) extracting surface meshes to anchor the Gaussians via barycentric coordinates, 3) establishing unsupervised semantic correspondences using GCN, 4) learning a neural morphing flow for non-linear interpolation of geometry and appearance. Experimental results on TexMorph show  improved performance of color consistency error, edge integrity over baseline methods, with qualitative improvements in handling non-isometric deformations and better user study preferences.

**Strengths:**

This paper integrates meshes with 3D Gaussians, enabling semantic-aware morphing that bridges the gap between unstructured point-based representations and structured topology.

The proposed method exhibits outstanding results on the TexMorph benchmark, achieving state-of-the-art performance.

The paper introduces novel benchmarks and metrics (i.e. TexMorph, MSE-SSIM, ∆E, EI) that advance evaluation standards for 3D morphing

**Weaknesses:**

The mesh-anchored Gaussian binding using barycentric coordinates and normal offsets is very similar to prior works like Dynamic Gaussians Mesh, which introduces Gaussian/Mesh Anchoring for aligning Gaussians to mesh faces in dynamic scenes, and Mesh-based Gaussian Splatting, which defines Gaussians over meshes for deformation. The deformation via neural morphing flow also seems to be studied in MaGS

-Dynamic Gaussians Mesh: Consistent Mesh Reconstruction from Dynamic Scenes Liu et al.

-Mesh-based Gaussian Splatting for Real-time Large-scale Deformation Gao et al.

-MaGS: Reconstructing and Simulating Dynamic 3D Objects
with Mesh-adsorbed Gaussian Splatting

Ablation studies are not sufficient. Ablation studies only test mesh guidance and geometric distortion loss separately, but ignores the impact of ARAP energy or smoothness loss.

The dual-domain optimization is incremental, combining standard losses (geodesic distortion, ARAP) from prior shape interpolation works like NeuroMorph and Spectral Meets Spatial, without substantial novel formulations. I think this contribution is somewhat marginal.

The color initialization by averaging Gaussian RGB assumes uniform lighting, which may introduce biases in real-world varying illumination.

Some important technical details are missing. For example, the Correspondence Morphing Flow (Ψ) is very vaguely described as a neural network without any network architecture detail; the parameters like KNN value or epsilon for geodesic distance approximation in smoothness loss are omitted.

**Questions:**

The paper uses SuGaR which is based on Poisson reconstruction for mesh extraction, so it assumes watertight surfaces. Thus, will it fail for fragmented objects or open surfaces? This leads to inconsistent anchoring.

Do you rigorously verify if “semantic correspondences” are established in an unsupervised manner? as claimed in the paper? Can we integrate 2D image priors to enhance robustness to non-isometric deformations and improve semantic accuracy?

Can we add a perceptual loss (e.g., LPIPS) for color consistency, capturing higher-level texture semantics and leading to more visually pleasing transitions?

---

> ### Author Response · Authors · 2025-11-21
> **Response to Reviewer ib2M(1/2)**
>
> We thank Reviewer ib2M for their constructive comments on our work.
> ### **Weakness 1: Regarding the Mesh-Gaussian method and the neural morphing flow**
> We thank the reviewer for raising this concern. While employing a mesh-Gaussian representation, GaussianMorphing uses a different scheme for barycentric coordinates from prior works.
> - **DG-Mesh**: Its "Anchoring" is a guiding and sampling process to ensure uniform point distribution for mesh extraction, distinct from a barycentric-based binding representation.
> - **MaGS**: Its "Adsorbed" representation serves reconstruction fidelity by dynamically learning coordinate changes ($\Delta w$).
> - **GaussianMorphing**: Our "Anchored" binding serves morphing quality by fixing barycentric coordinates to ensure cohesive texture-geometry transformation.
>
> About the Neural Morphing Flow :
> | Feature | GaussianMorphing (Ours) | MaGS |
> | :--- | :--- | :--- |
> | **Core Task** | **Inter-Object Semantic Morphing** Generates a path from Object A $\to$ Object B | **Intra-Object Dynamic Reconstruction & Simulation** Reconstructs a path from Object A(t=1) $\to$ A(t=2) |
> | **Network Function** | **Generator** The "Neural Morphing Flow" ($\Psi$) generates a new, non-linear interpolation path between two distinct objects | **Reconstructor / Simulator** Networks (MPE-Net, RMD-Net, RGD-Net) learn and reconstruct the intrinsic motion of a single object from video |
> | **Network Input** | **Semantic Correspondence Matrix ($\Pi$)** An inter-object mapping (learned via GCN) that guides the morphing | **Pose Information** An intra-object pose embedding (extracted via MPE-Net) that guides reconstruction |
> | **Primary Loss** | **Shape Matching & Interpolation Losses**  Ensure the generated path is geometrically and semantically plausible | **Rendering Reconstruction Loss**  Ensure the rendered image matches the ground-truth video frame |
>
> ---
>
> ### **Weakness 2: About the Ablation studies**
> Thank you for your valuable suggestion. We have updated **Sec 4.3** of the revised paper to include specific ablation experiments for the ARAP loss. Please refer to the newly added **Fig 8** and **Fig 9** for the corresponding analyses.
>
> ---
> ### **Weakness 3: About the assumes uniform lighting**
> We clarify that evaluating SH from a canonical viewing direction is not an assumption of uniform scene lighting. Rather, it is a strategy to "bake" view-dependent Gaussian appearance into consistent vertex attributes strictly for initialization purposes. Crucially, this serves only as a starting point. The actual visual transition is governed by the **appearance consistency loss** (${L}_{smooth}$), which ensures seamless morphing by enforcing local color smoothness weighted by geodesic distance. While this design prioritizes texture coherence over modeling complex dynamic illumination, it effectively prevents artifacts during the morphing process.
>
> ---
>
> ### **Weakness 4: Some important technical details are missing**
> We thank the reviewer for this feedback and have updated the paper accordingly.
> - Correspondence Morphing Flow $\Psi$ (**Sec 3.2**) : We have clarified that the Morphing Flow network utilizes a Graph Convolutional Network (GCN) architecture, sharing the same structural design as our feature extractor. Crucially, it accepts the time step $t$ as an additional input condition to learn the continuous non-linear interpolation flow defined in Eq. 3.
> - Hyperparameters: We have added the specific values for the KNN graph construction (**Appendix A.2**) and the smoothness loss epsilon (**Sec 3.3**) to the revised manuscript.

---

> ### Author Response · Authors · 2025-11-21
> **Response to Reviewer ib2M(2/2)**
>
> ### **Q1: Regarding the Inconsistent Anchoring from Watertight Assumption**
> We acknowledge that Poisson reconstruction generally favors watertight surfaces. However, our implementation leverages the FrostingGaussian framework, which explicitly incorporates regularization terms to ensure the extracted mesh accurately reflects the geometry captured by the 3D Gaussians.This strong geometric adherence mitigates potential anchoring inconsistencies.
>
> While we recognize that this approach may be suboptimal for severely fragmented or multi-component inputs, it proves sufficiently robust for the single, connected objects that constitute the primary scope of this work. In these scenarios, the regularized scaffold effectively ensures consistent and high-fidelity morphing.
>
> ---
>
> ### **Q2: Clarification on Unsupervised Semantic Correspondences**
> We confirm that our semantic correspondences are established in an **unsupervised manner**, requiring no labeled data. The alignment is learned by optimizing a GCN guided purely by intrinsic geometric priors (${L}_{geo}$) and the mesh topology, not by explicit semantic labels. This is also verified by the model's ability to generalize: a network trained on one pair can be successfully applied to new, unseen instances of the same category (e.g., the animal examples in Fig 3).
>
> We fully agree that integrating 2D image priors is a promising direction to enhance robustness against the significant topological changes. We are actively exploring how 2D semantic constraints can help relax the strict topological requirements of the mesh scaffold, using 2D guidance to ensure structural continuity and stability in challenging morphing scenarios.
>
> ---
> ### **Q3: About perceptual loss for color consistency**
> We thank the reviewer for this insightful question，We determined that reference-based 2D metrics like LPIPS are ill-suited for this task for two key reasons:
> - **Lack of Supervision**: Morphing is an unsupervised interpolation task; thus, no ground truth exists for intermediate frames, rendering supervised losses inapplicable.
> - **Sensitivity to Misalignment**: LPIPS incorrectly penalizes spatial misalignment, confounding valid geometric deformations with perceptual error.
>
> Instead, we employ our 3D-native appearance consistency loss (${L}_{smooth}$). By operating directly on the mesh manifold using geodesic weighting, this intrinsic constraint ensures coherent appearance transitions without relying on undefined 2D references.

---

### Official Review · Reviewer_oGM6 · 2025-10-25

**Soundness:** 2
**Presentation:** 3
**Contribution:** 2
**Rating:** 4
**Confidence:** 3

**Summary:**

The paper introduces a framework for 3D shape and texture morphing from multi-view images. The approach leverages mesh-guided 3D Gaussian Splatting (3DGS) to overcome the limitations of previous methods that rely on point clouds or pre-defined homeomorphic mappings. The core idea is a unified deformation strategy that anchors 3D Gaussians to reconstructed mesh patches, ensuring geometrically consistent transformations and preserving texture fidelity through topology-aware constraints. The framework also establishes unsupervised semantic correspondence using mesh topology and maintains structural integrity via physically plausible point trajectories. The method is evaluated on a new benchmark called TexMorph and shows improvements over existing 2D/3D morphing techniques.

**Strengths:**

1. The paper presents a new approach to 3D morphing by combining mesh-guided deformation with 3D Gaussian Splatting.

2. The method incorporates semantic awareness by using mesh topology as a geometric prior, enabling more meaningful and coherent morphing results.

3. The method requires only multi-view images as input, reducing the need for high-quality 3D data or manual annotations.

4. The paper introduces a new benchmark (TexMorph) and evaluation metrics tailored for 3D morphing, allowing for a more thorough assessment of the proposed method.

**Weaknesses:**

1. The smoothness requirement in the loss function, particularly the "Appearance Consistency" term (Lsmooth), may lead to over-smoothed textures and loss of fine details. The paper acknowledges this to some extent, but it remains a significant concern.

2. While the method reduces the need for specialized 3D assets, the computational cost of generating the initial mesh-Gaussian representation and optimizing the morphing framework is relatively high, requiring significant GPU resources and time.

3. Although the method introduces several metrics, the qualitative results do not look that good.

**Questions:**

-How does the method handle significant topological changes during morphing (e.g., when objects merge or split)?

-What are the limitations of using a mesh as a guiding structure? Are there scenarios where the mesh might hinder the morphing process?

-How well does the method generalize to datasets with different characteristics than TexMorph? （e.g. with highly structured and detailed texture/appearance)

---

> ### Author Response · Authors · 2025-11-21
> **Response to Reviewer oGM6(1/2)**
>
> We are grateful to the reviewer for their thorough evaluation and insightful comments.
>
> ### **Weakness 1: About the Appearance Consistency**
> We emphasize that the appearance consistency loss (${L}_{smooth}$)  plays a pivotal role in preventing color bleeding and ensuring coherent texture transitions, particularly given the **unsupervised** nature of our framework which lacks intermediate supervision.
>
> To explicitly prevent over-smoothing, we employ an adaptive regulation mechanism via geodesic distance-based inverse weighting (Eq. 7). This design selectively enforces smoothness within local neighborhoods while preserving sharp details across efficiently distant regions.
>
> Quantitatively, our method achieves a SOTA ${\Delta E}$ metric of 6.40 (significantly lower than baselines at 8.23–105), which validates it necessity and effectiveness in balancing smoothness constraints with detail preservation.
>
> ---
>
> ### **Weakness 2: Regarding the Initial Setup and Training Cost**
> We acknowledge the initial computational investment required for representation generation and framework training. However, is essential for establishing the high fidelity, structural coherence of our method. We emphasize that this cost is effectively amortized during deployment:
>
> our network is designed for robust category-level generalization. Once trained, the method can morph new, topologically similar pairs (as demonstrated by our animal examples in Fig 3) via a single forward pass, eliminating the need for re-optimization and enabling highly efficient, high-quality generation for practical applications.
>
> ---
>
> ### **Weakness 3：About the several metrics and the qualitative results**
>
> We thank the reviewer for insightful comments. While qualitative perception can be subjective, our method achieves a rigorous balance between geometric stability and texture fidelity. This is corroborated by our state-of-the-art quantitative metrics:
>
> | Metric | Purpose and Core Contribution | Quantitative Superiority (Ours vs. SOTA) |
> | :--- | :--- | :--- |
> | **Edge Integrity (EI)** | Measures Structural Stability and Silhouette Coherence, validating the effectiveness of our topology-aware constraints. | **9.0** (vs. NeuroMorph 13.0), indicating significantly reduced edge fragmentation. |
> | **Color Consistency(${\Delta E}$)** | Measures Texture Coherence in the unsupervised morphing trajectory, crucial for preventing color bleeding. | **6.40** (vs. MorphFlow 8.23), ensuring minimal color bleeding. |
> | **MSE(SSIM)** | Measures Temporal Geometric Consistency, quantifying the stability of the entire morphing sequence. | **0.11** (vs. MorphFlow 0.17), reflecting the fewest structural artifacts. |
>
> This perceptual superiority is validated by a user study ($N=54$), where $>80\%$ of participants preferred our method across all metrics.

---

> ### Author Response · Authors · 2025-11-21
> **Response to Reviewer oGM6(2/2)**
>
> ### **Q1: Regarding to handle the significant topological changes**
> We thank the reviewer for the question. The proposed framework is designed to establish semantically meaningful correspondences between objects sharing similar topologies. By learning a continuous correspondence map $\Pi$ on a mesh scaffold, the method explicitly enforces topological homeomorphism. Furthermore, the topology-aware losses are formulated to preserve this structural integrity rather than induce topological alterations.
>
> Consequently, handling fundamental topological events such as "merging" or "splitting" (genus changes) falls outside our current scope. We note that this is a shared characteristic of established 3D-centric geometric morphing methods; leading approaches like **NeuroMorph** (Eisenberger et al.,) and **Spectral Meets Spatial** (Cao et al.,) similarly prioritize topology-preserving matching over topological transitions. We acknowledge this as a limitation and an exciting direction for future research.
>
> ---
>
> ### **Q2: The limitations of using a mesh as a guiding structure**
>
> While meshes provide essential topological structure for guiding deformations and ensuring continuity, their utility is fundamentally constrained by the **fixed connectivity inherent to mesh representations**. This static nature poses a challenge when handling significant topological changes (e.g., morphing an apple into a donut), as edge relationships cannot be dynamically reconfigured during the deformation process.
>
> These limitations are an accepted trade-off to ensure the structural coherence and stability (Fig 6) that our mesh-guided approach provides across the entire morphing sequence.
>
> ---
>
> ### **Q3: Regarding Generalization to  Datasets**
>
> Thank you for highlighting this important aspect. Addressing the scarcity of morphing datasets with **high-fidelity, detailed textures** was indeed a primary motivation for proposing our TexMorph benchmark.
>
> Our experiments validate the method's robustness to such complex characteristics. For instance, the "dog→lion" and "cow→giraffe" transformations (Fig. 3) demonstrate our robust handling of fine, highly structured pattern details. Furthermore, we explicitly demonstrate generalization to diverse, non-synthetic domains, achieving high-quality results on real-world 3D scans (GSO dataset ) and in-the-wild mobile captures (Fig. 4)

---

> > ### Comment · Reviewer_oGM6 · 2025-11-28
> >
> > Thanks for the authors' clarification. Some of the details are clear now.
> >
> > However, after reading the authors' rebuttal and other reviewers' original comments, I still concern about the capability and scope of this paper. It claims that the approach deals with objects that share similar topologies. In fact, the original Gaussian Splatting representation was not limited to any topological structure, which has the potential to be morphed into any other structurally different objects, as shown in a lot of 3DGS-editing works. The limitation of dealing with topology differences comes from the introduced mesh-based pipeline and mesh-based correspondence estimation. The contradictory aspects of this method and the lack of potential to extend it to more sophisticated morphing cases, make me still lean negative towards this paper.

---

> > > ### Author Response · Authors · 2025-11-28
> > >
> > > We appreciate the reviewer’s accurate observation regarding the topological nature of 3DGS.
> > >
> > > We agree that 3DGS is not topologically limited by fixed connectivity. However, this unstructured nature is a double-edged sword. Without structural guidance, morphing 3DGS into structurally different objects is highly prone to local minima. This often leads to **structural disintegration**, manifesting as severe geometric inconsistencies and **shattered artifacts** (as shown in **Fig 6**).
> > >
> > > It is important to distinguish between *shape morphing* and *texture editing*. Most existing 3DGS editing works prioritize appearance synthesis or limited geometric deformations (e.g., bending), which rely on a stable underlying geometry. Fundamentally, these methods imply **identity invariance**—focusing on modifying attributes of the **same object** rather than handling the drastic topological changes required for morphing.
> > >
> > > Our **mesh-bounded strategy** fills this gap by handling the geometric challenge. By binding Gaussians to a mesh proxy, we ensure the morphing process remains structurally valid. Crucially, our method is **orthogonal** to texture-editing approaches. Since we retain the standard Gaussian representation, we can directly leverage existing SOTA texture-editing techniques to refine the appearance of our morphed results, combining our robust geometry with their high-quality texture synthesis.

---

### Official Review · Reviewer_sqZz · 2025-10-31

**Soundness:** 2
**Presentation:** 3
**Contribution:** 3
**Rating:** 4
**Confidence:** 4

**Summary:**

This paper introduces GaussianMorphing, a novel framework that unifies 3D geometry and texture morphing through a mesh-guided 3D Gaussian Splatting (3DGS) representation. The method anchors unstructured Gaussians to reconstructed mesh faces, integrating geometric and texture consistency via geodesic distortion and color-smoothness losses, achieving the balance between image-based pipelines and 3D-centric methods

**Strengths:**

1.The manuscript is well organized and easy to follow. The formulas and methodological details are clearly presented, making the technical contributions easy to understand.

2.The first-frame editing pipeline is a well-established paradigm in video editing, and I appreciate the exploration of extending this idea to 3D editing.

3.The presented editing results are promising and appear to be on par with the state-of-the-art performance of current 3D editing methods.

**Weaknesses:**

My concerning mainly lies on the setting. In my understanding, the main contribution of this paper is the view sampling issue. In another world, after optimally selecting the novel views, the rest processing is feed-forwarding the views to current 3D editing models. Current experiments seem to focus more on the comparison between the results with view expansion and that without view expansion to emphasis the effect of view expansion. In my opinion, the idea of
using view expansion itself is not sufficiently novel, as it is a common sense in reconstruction. Instead, I would like to see the improvement of proposed view sampling strategy corresponding to the baseline random sampling or uniform sampling, presenting how and why the proposed sampling strategy outperforms a vanilla strategy. I would temporarily give a borderline reject rating and raise my score upon this contribution is well illustrated.

**Questions:**

See weaknesses. I would suggest the authors respond to the concern.

---

> ### Author Response · Authors · 2025-11-21
> **Response to Reviewer sqZz**
>
> We sincerely thank the Reviewer for their time and effort in evaluating our submission.
>
> However, we respectfully note that the comments appear to address a paper with a substantially different core focus and methodology than our own. We suspect this may be due to a technical mix-up during the review upload process.
>
> To assist in clarification, our work, GaussianMorphing, introduces a novel framework for semantic-aware 3D shape and texture morphing from multi-view images. Our core contribution is a hybrid representation that leverages mesh-guided 3D Gaussian Splatting (3DGS). By anchoring the high-fidelity 3DGS appearance model to an explicit mesh skeleton, we ensure both structural stability and geometric consistency, effectively reconciling the inherent trade-off between stable geometry and high-fidelity appearance found in prior methods.
>
> We would firmly appreciate it if you could verify the review content. We look forward to receiving your updated feedback specifically regarding our work.

---

### Official Review · Reviewer_3AFn · 2025-11-01

**Soundness:** 3
**Presentation:** 3
**Contribution:** 3
**Rating:** 8
**Confidence:** 4

**Summary:**

The paper introduces GaussianMorphing, a novel framework for semantic-aware 3D shape and texture morphing that generates high-fidelity 3D outputs directly from multi-view images.

The core of the method is a hybrid paradigm that anchors unstructured 3D Gaussians to reconstructed mesh patches, utilizing the explicit mesh topology as a scaffold to guide geometrically consistent transformations while preserving texture fidelity. To handle complex transformations, the framework establishes unsupervised semantic correspondence using a Graph Convolutional Network (GCN) to capture local geometric context, thereby eliminating the need for labeled data or pre-aligned 3D assets. The process is governed by a dual-domain optimization strategy that employs geodesic-aware geometric distortion constraints ($L_{geo}$) and a texture-aware color smoothness loss ($L_{smooth}$) to ensure stable, structurally sound, and visually seamless transitions.

Through comprehensive experiments on the new TexMorph benchmark, GaussianMorphing substantially outperforms prior 2D and 3D methods, demonstrating superior structural consistency and texture preservation.

**Strengths:**

- GaussianMorphing introduces a novel hybrid paradigm that integrates 3D Gaussian Splatting (3DGS) with mesh-guided deformation. By using the mesh as a topological scaffold to anchor unstructured Gaussians, the method enables geometrically consistent transformations while preserving high-fidelity texture and appearance.
- GaussianMorphing achieves state-of-the-art performance on the newly proposed TexMorph benchmark, substantially outperforming existing 2D and 3D techniques. The approach demonstrates robust generalization across diverse sources, including complex synthetic models, real-world scanned objects, and in-the-wild photographs.
- The paper proposes unsupervised semantic correspondence and a dual-domain optimization strategy that combines geodesic-aware geometric distortion constraints with texture-aware color smoothness, ensuring stable, structurally sound, and visually seamless transitions.

**Weaknesses:**

While the final generation of the morphing sequence is fast (around 2 minutes), the initial setup and training phase suggest high computational demand. Generating the initial hybrid mesh-Gaussian representation takes about 1 hour for a typical object pair, followed by optimization requiring 500 to 1000 iterations. This multi-stage process indicates that the time and resource cost to enable morphing between a new pair of objects is still relatively high compared to some image-based methods (e.g., FreeMorph is tuning-free).

**Questions:**

- Does the current method require the source and target to be topologically homeomorphic when establishing correspondences? For example, would transformations like "apple to donut" pose inherent difficulties? If such cases can be accommodated, could the revision include corresponding qualitative and quantitative results?

- Does the network for extracting semantic-aware mesh correspondences require pretraining, or is it trained jointly with the morphing flow network?

- If providing the full code is not feasible, pseudocode would be sufficient to clarify the method. Including incomplete code in the supplementary material to suggest actual training code may cause confusion and is not needed.


- Yang et al., 2025: "Textured 3D regenerative morphing with 3D diffusion prior" has goals closely aligned with GaussianMorphing, namely achieving textured 3D morphing using 3D priors. As this work is closer in paradigm, it would be helpful to include more discussion.

- Does the method risk collapsing intermediate transformations to a trivial state before reaching the target? For example, could the process degenerate from the source to a gray spherical Gaussian representation (which might serve as a generic initialization) and then to the target? If this does not occur, it would be better to explain why the method avoids such collapse. In cases where the source and target are highly dissimilar in geometry and texture, such as morphing from a plant to an animal, does a similar issue arise?

---

> ### Author Response · Authors · 2025-11-21
> **Response to Reviewer 3AFn(1/2)**
>
> We express our sincere gratitude to the reviewer for the insightful comments and the recognition of our work. We have addressed the specific concerns raised by the reviewer as detailed below.
>
> ### **Weakness: Regarding the Initial Setup and Training Cost**
>
> We wish to clarify that the optimization overhead is not incurred on a per-pair basis for in-category morphing. By leveraging a generalizable category-level network, our method obviates the need for pair-specific optimization (i.e., the 500–1000 iterations) for novel intra-category pairs. Consequently, following one-time training, GaussianMorphing achieves highly efficient inference via a single forward pass.
>
> ---
>
> ### **Q1: Topological Homeomorphism and Applicability to Non-Homeomorphic Morphing**
>
> Our work explicitly focuses on **semantic-aware morphing**, primarily targeting intra-category objects or those sharing similar semantic structures. To achieve precise geometry-texture alignment, we utilize a mesh-guided strategy that relies on **topological homeomorphism**. We acknowledge that handling morphing with fundamental topological differences (e.g., "apple to donut") is a current limitation. Such transformations often lack clear semantic correspondence (e.g., a "hole"), and our model struggles to plausibly create such new structures.
>
> We consider this problem of topology-changing morphing to be a distinct and challenging research direction for the community.
>
> ---
>
> ### **Q2: Regarding the Semantic Correspondence Network Training**
>
> We clarify that the networks responsible for extracting semantic correspondence ($\mathbf{\Pi}$) and predicting the morphing flow ($\Psi$) are tightly coupled and jointly optimized.
>
> While sharing the same GCN architecture, $\mathbf{\Pi}$ is dynamically learned and refined throughout the morphing flow, rather than being fixed beforehand. The geometric losses of adjacent frames, driven by the intermediate states of the morphing flow $\Psi(t)$, serve as a feedback mechanism to optimize the learned correspondence.

---

> ### Author Response · Authors · 2025-11-21
> **Response to Reviewer 3AFn(2/2)**
>
> ### **Q3: Pseudocode is Sufficient**
>
> We have updated the Appendix (A.4) with detailed pseudocode to further clarify the algorithmic flow of our method, and the full implementation will be released publicly in the future.
>
> ---
>
> ### **Q4: Detailed Comparison with Texture3D**
>
> We appreciate the reviewer's suggestion. We have now incorporated a more detailed introduction and analysis of the Texture3D approach within the Related Work section of the revised paper (Line 139).
>
> ---
>
> ### **Q5: Preventing Degeneration in Intermediate Morphing States**
>
> We clarify that our method inherently prevents intermediate collapse through two mechanisms:
>
> * The **explicit mesh skeleton** provides a consistent, preserved topology, which rigidly constrains the deformation space of the anchored 3D Gaussians.
> * The **geometric consistency losses**  ($L_{arap}, L_{geo}$) operate between consecutive frames to preserve local shape coherence and actively prevent structural drift toward a trivial state.
>
> Consequently, even in highly dissimilar cases (e.g., "Plant to Animal"), the failure mode is semantic misalignment rather than structural degeneration.

---

### Author Response · Authors · 2025-11-21
**General Response**

We sincerely thank all the reviewers for their insightful reviews and constructive feedback. We wish to clarify a primary misconception regarding the training efficiency.

**GaussianMorphing is fundamentally designed as a generalizable category-level network.** While our framework retains the flexibility to perform **pair-specific optimization** for new objects to maximize fidelity, this is **optional**. By leveraging a learned category-prior, our method obviates the need for iterative optimization for novel intra-category pairs. Consequently, following **one-time training**, we achieve highly efficient inference via a **single forward pass**. This distinct design ensures both robust, instant morphing for general use and the potential for fine-grained one-to-one optimization when desired.

Addressing the concern on **topological scope (Reviewer oGM6)**, we emphasize that while 3DGS is unstructured, this flexibility often leads to **local minima** and **structural disintegration** during drastic morphing (Fig. 6). We distinguish our work from existing 3DGS-editing methods, which primarily focus on **texture editing** or limited deformation under **identity invariance** (modifying attributes of the same object). Our **mesh-bounded strategy** addresses the geometric challenge of **shape morphing** by ensuring structural validity, making our method **orthogonal** to and compatible with appearance-focused editing techniques.

In addition to this clarification, we have revised our manuscript to incorporate your valuable suggestions and provide further technical details. The key updates are summarized below:

* **Clarification on Network Architecture:** We have clarified in Sec 3.2 (Line 222) that the Morphing Flow network ($\Psi$) utilizes a GCN architecture analogous to our feature extractor and accepts the time step $t$ as an additional input to learn non-linear interpolation.
* **Hyperparameters Details:** We have revised the paper to include specific hyperparameter values, such as the KNN settings in Appendix A.2 (Line 698) and the Epsilon value in the smoothness loss in Sec 3.3 (Line 268).
* **Expanded Related Work:** We incorporated a detailed introduction and comparative analysis of the Texture3D approach within the Related Work section (Line 139).
* **Updated Ablation Studies:** We have updated the ablation experiments regarding the ARAP constraint in Sec 4.3, specifically referencing Figure 8 and Figure 9.
* **Pseudocode Refinement:** We have updated the detailed pseudocode in Appendix A.4 to further clarify our method's implementation.

---

### Meta-Review · Area_Chair_pacB · 2026-01-06

**Summary:**

The paper presents GaussianMorphing, which morph between two 3D objects given as multiview images. They first reconstruct 3D-GS anchored meshes of the objects, and then train graph counvolutional network to learn the correspondence and the deformation flow. The paper initially received one 8 rating and three 4 ratings. One of the 4 rating review is irrelevant to the current work and is negelected for the final decision. The reviewers acknowledged GaussianMorphing delivers more meaningful and coherent morphing results. The main raised concerns are 1) the dependency on 3D mesh/mesh quality; 2) novelty of the proposed method. In particular, 1) the method's success is heavily dependent on the quality of the mesh extracted from 3DGS. If the mesh construction fails, the morphing would fail. It further has topology limitations by relying on 3D mesh; 2) The technical novelty is also limited. The proposed method is largely assembled from prior existing works (i.e. SuGaR-CVPR 2024, and NeuroMorph-CVPR 2021). Regrettably, the AC cannot recommend acceptance for this paper after carefully weighing the paper's strengths against its limitations.

**Reviewer Concerns:**

The AC think the concerns have been clarified by the rebuttal.

**Reviewer Scores:**

The AC think the reviewers who give negative ratings might not upgrade their scores. It is also mentioned by oGM6 that "The contradictory aspects of this method and the lack of potential to extend it to more sophisticated morphing cases, make me still lean negative towards this paper."

---

### Decision · Program_Chairs · 2026-01-26

Reject